# The PTS Components in *Klebsiella pneumoniae* Affect Bacterial Capsular Polysaccharide Production and Macrophage Phagocytosis Resistance

**DOI:** 10.3390/microorganisms9020335

**Published:** 2021-02-08

**Authors:** Novaria Sari Dewi Panjaitan, Yu-Tze Horng, Chih-Ching Chien, Hung-Chi Yang, Ren-In You, Po-Chi Soo

**Affiliations:** 1Department of Laboratory Medicine and Biotechnology, College of Medicine, Tzu Chi University, No. 701, Sec. 3, Zhongyang Rd., Hualien 97004, Taiwan; novariasari.p@gmail.com (N.S.D.P.); d92424001@ntu.edu.tw (Y.-T.H.); yri100@mail.tcu.edu.tw (R.-I.Y.); 2Graduate School of Biotechnology and Bioengineering, Yuan Ze University, Taoyuan 32003, Taiwan; ccchien@saturn.yzu.edu.tw; 3Department of Medical Laboratory Science and Biotechnology, Yuanpei University of Medical Technology, Hsinchu 30015, Taiwan; hcyang@mail.ypu.edu.tw

**Keywords:** bacterial physiology, crr deletion mutant, capsular polysaccharide

## Abstract

Capsular polysaccharide (CPS) is a crucial virulence factor for *Klebsiella pneumoniae* infection. We demonstrated an association of CPS production with two phosphoenolpyruvate:carbohydrate phosphotransferase systems (PTSs). Deficiency of *crr*, encoding enzyme IIA of PTS, in *K. pneumoniae* enhanced the transcriptional activities of *galF*, *wzi* and *gnd*, which are in the *cps* gene cluster, leading to high CPS production. A *crr* mutant exhibited a higher survival rate in 1% hydrogen peroxide than the wild-type. The *crr* mutant showed less sensitivity to engulfment by macrophage (RAW 264.7) than the wild-type by observing the intracellular bacteria using confocal laser scanning microscopy (CLSM) and by calculating the colony-forming units (CFU) of intracellular bacteria. After long-term incubation, the survival rate of the intracellular *crr* mutant was higher than that of the wild-type. Deficiency of *crr* enhanced the transcriptional activities of *etcABC* which encodes another putative enzyme II complex of a PTS. Deletion of *etcABC* in the *crr* mutant reduced CPS production and the transcriptional activities of *galF* compared to those of the *crr* mutant. These results indicated that one PTS component, Crr, represses CPS production by repressing another PTS component, EtcABC, in *K. pneumoniae*. In addition, PTS plays a role in bacterial resistance to macrophage phagocytosis.

## 1. Introduction

*Klebsiella pneumoniae*, a member of the *Enterobacteriasceae* family, is an opportunistic pathogen that generally causes pneumonia, urinary tract infection (UTI) and bacteremia in hospitalized patients or patients with underlying diseases, such as diabetes mellitus, renal impairment and chronic liver disease [1,2,3]. The habitats for *K. pneumoniae* are ubiquitous and include humans, animals, plants, natural surface waters, and soils [4,5,6]. *K. pneumoniae* is reported to be a member of the gut microbiota in humans and animals, such as cows, pigs, birds, fish, house flies, and earthworms [6,7,8,9,10,11,12]. The prevalence of *K. pneumoniae* in animals differs among studies. For example, *K. pneumoniae* was detected in 32–92% of cattle [6,13]. However, the rate in humans may be much lower. *K. pneumoniae* could be found in 3.8% of stool samples and 9.5% of nares samples from healthy volunteers or 6% of rectal and throat swab samples from patients in the intensive care unit (ICU). In addition, *K. pneumoniae* in a patient’s own microbiota is supposed to play a key role in *K. pneumoniae* infections [10,14]. The well-studied virulence factors of *K. pneumoniae* assisting in survival within the host are capsular polysaccharide (CPS), lipopolysaccharide (LPS), siderophores, and fimbriae (also known as pili).

Capsules are voluminous polysaccharide matrices surrounding bacterial cells [1,15]. *K. pneumoniae* strains with hypercapsules show hypermucoviscous colonies on agar plates. This hypermucoviscous phenotype can be semiquantified by the string test [16]. CPS protects bacteria from host immune defense. The role of CPS in *K. pneumoniae* anti-immune strategies involves several pathways described below. First, CPS assists bacteria in escaping complement binding and opsonization. Second, CPS reduces the bactericidal effects of antimicrobial peptides. Third, CPS protects bacteria from recognition and adhesion by phagocytes and epithelial cells [1,15,17]. In addition, CPS may protect prokaryotic and eukaryotic organisms from oxidative stress. Doyle et al. reported that *Streptococcus pneumoniae* treated with antibody against CPS was more sensitive to hydrogen peroxide [18]. Capsule enlargement protected *Cryptococcus neoformans* from the reactive oxygen species produced by hydrogen peroxide [19]. Furthermore, CPS also alters the activation of cytokines in the immune system cells or cell surface markers on epithelial cells [20]. Various kinds of phagocytes and epithelial cells have been experimentally infected by *K. pneumoniae* without or overproducing CPS. CPS-deficient mutants are more vulnerable to ingestion by MH-S mouse alveolar macrophages than wild-type *K. pneumoniae* [21]. The binding of CPS-deficient mutants to human monocyte-derived dendritic cells (mo-DCs) is greater than that of wild-type *K. pneumoniae* [22]. A *K. pneumoniae* strain with a large quantity of CPS was significantly more resistant to phagocytosis by neutrophils than strains with a small quantity/lack of CPS [23]. The internalization of a CPS-deficient mutant by human lung carcinoma cells (A549) was more efficient than that of wild-type bacteria. In addition, an inverse relationship was observed between the amount of CPS produced by a strain and the amount of bacteria inside A549 cells [24]. The genes for CPS production are clustered in the chromosome of *K. pneumoniae* (*cps* cluster). In most capsular types, the highly conserved genes are *galF*, *cpsACP, wzi, wza, wzb*, and *wzc* at the 5′ end of the *cps* locus, which are involved in the translocation and surface assembly of CPS, while *gnd* and *ugd* at the 3′ end of the *cps* locus encode glucose-6-phosphate dehydrogenase and UDP-glucose dehydrogenase, respectively. Genes between *wzc* and *gnd* are variable among different capsular types [25].

Bacteria have several phosphoenolpyruvate (PEP): carbohydrate phosphotransferase systems (PTSs) to translocate and phosphorylate various sugars. In general, the PTS contains enzyme I (EI), histidine-containing phosphocarrier protein (HPr) and enzyme II (EII) complexes. EI and HPr are cytoplasmic proteins and are common to all PTSs, while EII complexes containing A, B and C (and sometimes D) proteins/domains are sugar specific and vary in each PTS. In addition to sugar transport, the components of PTSs have been reported to be involved in various bacterial physiological processes. For example, Crr, located in the cytoplasm, is a glucose-specific EIIA (EIIA^Glc^) of the PTS in enteric bacteria [26]. Dephosphorylated Crr was demonstrated to interact with glycerol kinase (GlpK) and the maltose/maltodextrin ABC transport system (MalK), leading to inhibition of glycerol and maltose transport in *Escherichia coli* and *Salmonella enterica* serovar Typhimurium. In *E. coli*, the dephosphorylated form of Crr also interacted with lactose permease (LacY) to prevent the uptake of lactose in the presence of glucose. In addition, the phosphorylated/dephosphorylated form of Crr bound adenylate cyclase (AC), leading to an increase/decrease in the synthesis of cAMP, respectively, in *E. coli*. Then, the cellular cAMP concentration affected many bacterial catabolic genes [27]. In *Vibrio vulnificus*, EIIA^Glc^ sequesters flagellar assembly protein A (FapA) from the flagellated pole to inhibit bacterial motility in the presence of glucose [28]. Oh et al. reported that a *crr* mutant strain displayed enhanced production of 1,3-propanediol from glycerol in *K. pneumoniae* compared to that in its parent strain, which was a *K. pneumoniae ldhA* and *wzi* double mutant derived from *K. pneumoniae* ATCC 200721 [29]. We previously reported that KPN00353 (EtcA homolog), KPN00352 (EtcB homolog) and KPN00351 (EtcC homolog) in *K. pneumoniae* are predicted to be EIIA, EIIB and EIIC homologs, respectively. *KPN00353*, *KPN00352* and *KPN00351* (homologous to *etcA*, *etcB* and *etcC*, respectively) are transcribed in an operon. By binding to glycerol kinase, KPN00353 (EtcA homolog) reduced bacterial glycerol uptake, leading to the decreased production of 1,3-propanediol from *K. pneumoniae* [30]. *K. pneumoniae* overexpressing KPN00353-KPN00352-KPN00351 (EtcABC homolog) produced more biofilm and CPS than the vector control [31]. We also found that the *K. pneumoniae* STU1 *crr* mutant strain lacked the ability to produce cAMP. However, overexpression of EtcABC could restore the cAMP level in the Δ*crr*/Δ*etcABC* double mutant and then positively regulate the type 3 fimbriae of *K. pneumoniae* STU1 [32].

In this study, we observed that the *K. pneumoniae crr* mutant showed hypermucoviscous colonies on agar plates. Then, the CPS amounts of the *crr* mutant and wild-type were quantified. By RT-qPCR, we found that the transcriptional activities of genes in the *cps* cluster were increased in the *crr* mutant compared to those in the wild-type. Infecting macrophages, we showed that the *crr* mutant was more resistant to phagocytosis than the wild-type. Subsequently, as determined by RT-qPCR, the mRNA levels of another EII complex, *etcABC*, in the *crr* mutant were greater than those in the wild-type. By gene deletion, we found that glucose-specific EIIA Crr played a role in the regulation of CPS production via the *etcABC*.

## 2. Materials and Methods

### 2.1. Bacterial Strains, Plasmids, and Growth Conditions

The bacterial strains, plasmids, and primers used in this study are listed in Appendix A. Unless otherwise stated, bacteria were routinely cultured on a rotatory shaker at 200 rpm and 37 °C in LB medium (10 g/L tryptone, 5 g/L yeast extract, and 10 g/L NaCl) supplemented with appropriate antibiotics at the following concentrations: ampicillin (100 µg/mL), chloramphenicol (100 µg/mL), gentamicin (20 µg/mL), penicillin (100 µg/mL), streptomycin (50 µg/mL) and kanamycin (50 µg/mL). Sheep blood agar plates (BAPs, 5%) were purchased from Dr. Plate Biotech Company (Taipei, Taiwan). The cells were cultivated using Dulbecco’s modified Eagle’s medium-high glucose (DMEM-glucose), which contains 4500 mg/L glucose and 4 mM L-glutamine, without sodium pyruvate, purchased from HyClone (Cytiva, USA). If the recombinant DNA in the pBSK-derivative plasmid or pBAD33-derivative plasmid was required to be expressed, the final concentration of isopropyl β-D-1-thiogalactopyranoside (IPTG) or arabinose was 50 μg/mL or 0.2% (*w/v*), respectively, in the medium.

### 2.2. Construction of the crr Mutant and Δcrr/ΔetcABC Double Mutant

The *crr* and *etcABC* genes were deleted using unmarked mutagenesis in *K. pneumoniae*. The approximately 1000-bp upstream and 700-bp downstream flanking DNA fragments of the target gene were amplified by PCR using the primers summarized in Appendix A. These two fragments were ligated into the suicide vector pW18mobsacB. The construction of pW18mobsacB, transfer of its recombinant plasmids into *K. pneumoniae* and confirmation of mutants were performed as described in a previous study [32]. The *etcABC* genes were deleted in the *crr* mutant, Δ*crr*, to obtain the Δ*crr*Δ*etcABC* double mutant.

### 2.3. String Test

The strains were inoculated on either LB agar or 5% sheep BAPs. After incubation at 37 °C overnight, the string test with a colony was performed according to the method reported by Fang et al. [33]. Hypermucoviscosity was defined when the strain formed a viscous string more than 5 mm long using a loop. The test was repeated at least three times for each strain.

### 2.4. Congo Red Agar (CRA) Assay

Bacteria were observed according to the method described by Rampelotto et al. with modification [34]. In brief, overnight bacterial cultures in LB were washed twice with phosphate-buffered saline (PBS) and then adjusted to an OD_600_ of 0.001. Subsequently, 2 μL of bacterial suspension was inoculated on brain heart infusion (BHI) (BD, USA) agar with the addition of 5% (*w/v*) sucrose and 0.8% (*w/v*) Congo red, followed by incubation at room temperature. Glycocalyx-producing bacteria formed black colonies, whereas non-glycocalyx-producing bacteria formed red colonies.

### 2.5. Bacterial Mucoviscosity Assay

The mucoviscosity levels were semiquantified according to the study by Wu et al., with modification [35]. The bacteria were cultured overnight and subcultured for another 6 h in LB broth at 37 °C with 200 rpm agitation. Subsequently, bacterial cells of each aliquot [OD_600_ × V(mL) = 1] were pelleted by centrifugation at 1500× *g* for 5 min. The mucoviscous bacteria were difficult to compact in a pellet after centrifugation. The absorbance of the supernatant was monitored at 600 nm using a spectrophotometer (Prema, Taipei, Taiwan).

### 2.6. Transmission Electron Microscopy (TEM)

After overnight culture in LB broth, 1 mL of bacterial suspension was centrifuged at 1500× *g* for 5 min. After being washed twice with PBS, the bacteria were resuspended and fixed in a primary fixation solution (2.5% glutaraldehyde, 0.1 M cacodylate buffer, and 1% tannic acid) at 4 °C for 1 h, followed by centrifugation at 1500× *g* for 5 min. Subsequently, the bacterial pellet was suspended in PBS. Ten microliters of bacterial suspension were absorbed onto 200-µm-pore-size mesh copper electron microscopy grids coated with carbon and Formvar. The excess liquid was removed, and the grids were rinsed twice with water. Then, the sample on the grid was negatively stained with a drop of 2% (*w/v*) uric acid for 15 s before observation by using a Hitachi H-7500 transmission electron microscope (Hitachi, Tokyo, Japan) operated under standard conditions with the cold trap in place.

### 2.7. Extraction and Quantification of Capsular Polysaccharide (CPS)

CPS extraction was performed as described by Horng et al., with some modifications. [31]. In brief, after overnight culture in LB, the bacterial cells were washed twice with 0.9% NaCl solution. Then, aliquot of bacterial cells [OD_600_ × V(mL) = 0.01] was incubated in 20 mL of CPG broth (1% tryptone peptone, 1% casamino acid, and 1% glucose) for 6 h. Subsequently, the bacterial concentration was detected by spectrophotometry at 600 nm (Prema, Taipei, Taiwan). Aliquots of bacterial cultures were collected according to the formula OD_600_ × volume (mL) = 5 and by centrifugation at 17,000× *g* for 30 min. The cell pellets were then suspended in 10 mL of high-salt buffer (10 mM potassium phosphate buffers, pH 7.0; 15 mM NaCl; 1 mM MgSO_4_) and mixed for 1 h at 4 °C by using a Vortex-Genie 2 (Scientific Industries, Bohemia, NY, USA) with the setting 5.0, followed by centrifugation at 17,000× *g* for 30 min. The CPS in the supernatant was precipitated by adding three volumes of ethanol and incubating at 20 °C overnight. After centrifugation at 17,000× *g* for 30 min, the CPS was suspended in 1 mL of water. The quantification of CPS was performed using the colorimetric method (phenol-sulfuric acid method), with some modifications [36]. In brief, 300 μL of the polysaccharide solution was mixed with 150 μL of phenol solution, pH 7.0, in the tube, followed by the slow addition of 750 μL of sulfuric acid along the side of the tube. After the tube was capped, the mixture was stirred vigorously by vortexing for 5 s. After incubation for 30 min at 80 °C, the absorbance of the solution was detected at a wavelength of 490 nm using a spectrophotometer (Prema, Taipei, Taiwan).

### 2.8. Reverse Transcription Quantitative Real-Time PCR (RT-qPCR)

The purified RNA was treated with 50 U/mL RNase-free DNase I (New England Biolabs, USA) for 30 min at 37 °C to remove residual genomic DNA. The mRNA was reverse transcribed using a QuantiNova reverse transcription kit according to the manufacturer’s instructions (Qiagen, Hilden, Germany). To quantify the cDNAs from the transcripts of *galF*, *wzi*, *gnd*, and *recA*, dye-based qPCR was performed in triplicate using the QuantiNova SYBR Green PCR Kit according to the manufacturer’s instructions (Qiagen, Hilden, Germany). To quantify the cDNAs from the transcripts of *galF* and *recA* or *etcA*, *etcB*, *etcC*, and *recA*, probe-based qPCR was performed in triplicate using fluorescein-labeled and dual-quenched probes (Integrated DNA Technologies, Coralville, IA, USA) (Appendix A) and TaKaRa Taq DNA polymerase (TaKaRa Bio, Shiga, Japan). All primers and probes for RT-qPCR are listed in Appendix A. Real-time PCR was performed by a Rotor-Gene real-time genetic analyzer (Qiagen, Hilden, Germany). The gene expression levels were normalized to those of 16S rRNA following the 2^−ΔΔCT^ method. The housekeeping gene *recA*, encoding recombinase A, was used as a reference [37].

### 2.9. Hydrogen Peroxide (H_2_O_2_) Sensitivity Assay

The assay was performed as previously described with modifications [38]. After overnight culture in LB medium at 37 °C with 200 rpm agitation, aliquots of bacterial cultures were adjusted according to the formula, OD_600_ × V(mL) = 1 × 10^−5^. Then, hydrogen peroxide was added to a final concentration of 1% (*v/v*) into 1 mL of bacterial culture, followed by incubation at room temperature for 30 min. Subsequently, the bacteria were serially diluted and spotted onto LB agar plates, followed by incubation overnight and calculation of the CFU.

### 2.10. Phagocytosis Assays (Macrophage Ingestion and Bacterial Survival Ability in Macrophages)

Phagocytosis of *K. pneumoniae* by the mouse macrophage cell line RAW 264.7 was performed by a previously described method, with some modifications [33]. Initially, cells were seeded at a density of 6 × 10^4^ per well in a 24-well plate (JET Biofil, Guangzhou, China). After washing with PBS, bacteria were added at an MOI = 100:1 for 1 h of incubation at 37 °C in DMEM-glucose media. Subsequently, the cells were washed twice with PBS and then treated with 500 μL of PBS containing 20 μg/mL gentamycin and 50 μg/mL streptomycin for 30 min to eliminate residual extracellular bacteria. After washing twice with PBS, aliquots of cells were added with DMEM-glucose medium and incubated for 2, 5 or 16 h to observe bacterial survival in the macrophages. The other aliquot of cells was immediately tested for macrophage ingestion. To assess the recovery of intracellular bacteria, the macrophages in the well were dissociated with trypsin solution and lysed with PBS containing 0.1% Triton X-100. Subsequently, intracellular bacteria were diluted and plated in triplicate onto LB agar plates. After 24 h of incubation at 37 °C, the resultant CFU were counted.

### 2.11. Phagocytosis Observed by Confocal Laser Scanning Microscopy (CLSM)

The mouse macrophage cell line RAW 264.7 was seeded at 6 × 10^4^ cells per well in a 24-well plate containing a 10-mm microscope cover glass (Glaswarenfabrik Karl Hecht, Sondheim vor der Rhön, Germany). The cells were infected with bacteria carrying pBSK-Km:ZsGreen in DMEM-glucose medium containing IPTG (0.5 mM) and kanamycin (50 µg/mL) for 1 h. After washing twice with PBS, the cells were incubated in PBS containing 20 μg/mL gentamycin and 50 μg/mL streptomycin for 30 min to eliminate residual extracellular bacteria. Subsequently, cells were fixed in 3% paraformaldehyde at room temperature for 30 min and then permeabilized with 0.1% Triton X-100 in PBS. After washing with PBS, the cells were counterstained with CF^®^647 phalloidin (Biotium, Fremont, CA, USA) to stain F-actin and DAPI (Invitrogen, Thermo Fisher Scientific, Waltham, MA, USA) to identify the nuclei. A 10-mm microscope cover glass with cells was placed and mounted on a microscope glass slide for observation by a Nikon C2Si laser confocal microscope configured with a Nikon Eclipse Ni upright microscope (Nikon, Tokyo, Japan) at a magnification of 600×. Confocal images were analyzed using 64-bit NIS-Elements AR version 4.60.00 imaging software (Nikon, Tokyo, Japan).

### 2.12. Statistical Analysis

For spectrophotometry, RT-qPCR, string length, capsule width, CPS quantification, CFU counting, and bacterial survival rate data, the values were expressed as the mean ± standard deviation from three independent bacterial cultures. Data were subjected to analysis of variance (ANOVA) and considered significantly different at *p* < 0.05.

## 3. Results

### 3.1. Phenotypic Characteristics of the crr Mutant in K. pneumoniae STU1

The role of Crr in *E. coli* has been studied well [26]. However, the role of *crr* in the pathogenesis of *K. pneumoniae* is unclear. As the *crr* mutant colonies looked more mucoviscous than the wild-type colonies on agar plates, we observed the *crr* mutant and wild-type by transmission electron microscopy (TEM). The results showed that the *crr* mutant displayed more extracellular matrix than the wild-type (Figure 1). Subsequently, we used the string test to determine whether the *crr* mutant was hypermucoviscous or nonhypermucoviscous. The average length of viscous strings formed by the wild-type, *crr* mutant and *crr* complemented strain was approximately 2.8, 8.1, and 3.6 cm, respectively, which was more than 5 mm, indicating that they were all hypermucoviscous strains. In addition, the length of viscous strings of the *crr* mutant was much longer than that of the wild-type. Compared to the *crr* complemented strain, the *crr* mutant carrying pBAD33 as a vector control also showed more extended viscous strings (Figure 2A and Appendix A). The mucoviscosity phenotype of the *crr* mutant could also be observed by low-speed centrifugation (mucoviscosity assay) because highly mucoviscous bacteria are difficult to pellet by low-speed centrifugation. After centrifugation, there were more suspended bacteria for the *crr* mutant than for the wild-type (Figure 2(Bi)). The OD_600_ of supernatant from the *crr* mutant sample was higher than that from the wild-type (Figure 2(Bii)). The *crr* complemented strain restored the wild-type phenotype (Figure 2B). Due to the positive string test and mucoviscosity assay results, we speculated that the *crr* mutant had more exopolysaccharide surrounding itself than the wild-type. As a polysaccharide capsule is an important virulence factor in *K. pneumoniae*, we examined capsule of *crr* mutant and wild-type.

### 3.2. Crr, an EIIA Component, Reduces CPS Synthesis in K. pneumoniae STU1

To observe the bacterial capsule, the *K. pneumoniae* STU1 wild-type and *crr* mutant were stained by India ink staining. The carbon particles of the ink poorly attached to the polysaccharides in the capsule, showing a clear halo surrounding the bacteria. Both the wild-type and *crr* mutant showed the surrounding capsule. Then, we measured the width of the rod-shaped bacteria to represent the capsule thickness. The *crr* mutant had a much thicker capsule than the wild-type and *crr* complemented strains (Figure 3A). In addition, Congo red is reported to be able to bind polysaccharides in a helical conformation to form complexes [39]. By Congo red agar (CRA) assay, slime-producing bacteria in a report by Freeman et al. and biofilm-producing bacteria in a report by Rampelotto et al. showed black colonies. By contrast, colonies of non-slime-/biofilm-producing isolates remained pink/red [34,40]. The *K. pneumoniae* STU1 wild-type and *crr* complemented strain on plates containing Congo red revealed gray colonies, while the *crr* mutant displayed black colonies, indicating that all of strains (wild-type, *crr* complemented strain and *crr* mutant) produced glycocalyx but the *crr* mutant may have more glycocalyx (Figure 3B). Therefore, we quantified the CPS produced by *K. pneumoniae* STU1 and its isogenic mutant. The results showed that the amount of CPS was increased in the *crr* mutant compared to that in the wild-type (Figure 3C). To further analyze whether the genes involved in CPS synthesis were affected by Crr, the transcriptional activities of *galF*, *wzi* and *gnd* in the wild-type and *crr* mutant were measured by RT-qPCR. Compared to those of *galF*, *wzi* and *gnd* in the wild-type, the relative transcription levels of these three genes in the *crr* mutant were higher (Figure 3D). In contrast, these genes in the *crr* complemented strain were decreased compared to those in the vector control (Figure 3E). The housekeeping gene *recA* was used as a reference [37]. The relative transcription levels of *recA* in the wild-type and *crr* mutant were similar (Figure 3D). These results indicated that Crr negatively affected *cps* locus transcription and CPS production.

### 3.3. Deficiency of crr Rendered Bacteria Resistant to Oxidative Stress and Phagocytosis

CPS was reported to protect yeast from the reactive oxygen species produced by hydrogen peroxide [19] and protect bacteria from ingestion by MH-S mouse alveolar macrophages [21]. Since the *K. pneumoniae crr* mutant highly produced CPS, the survival percentage of bacteria after 30 min of 1% hydrogen peroxide treatment was examined. The results showed that the *K. pneumoniae* wild-type and *crr* complemented strains were sensitive to hydrogen peroxide. In contrast, the *crr* mutant exhibited a high survival rate in the presence of hydrogen peroxide (Figure 4A). Then, the bacteria ingested by macrophages (RAW 264.7) were quantified by calculating the bacterial colony-forming units (CFU) and observed by confocal laser scanning microscopy (CLSM) (Figure 4B,D). After 1 h of exposure to macrophages and 30 min of antibiotic treatment, the number of CFU of the *crr* mutant engulfed by macrophages was significantly lower than that of wild-type *K. pneumoniae* ingested by macrophages (Figure 4B). To observe the fate of intracellular *K. pneumoniae*, RAW 264.7 cells were infected with bacteria expressing ZsGreen. By CLSM, ZsGreen-expressing wild-type bacteria and *crr* mutant could be observed inside macrophage cells after 1 h of exposure to macrophages and 30 min of antibiotic treatment. However, the average number of intracellular *crr* mutant was less than that of wild-type bacteria in macrophage cells (Figure 4D). The results indicated that the *crr* mutant was less vulnerable to macrophage ingestion than the wild-type. Moreover, the bacterial survival rate in the phagosomes of macrophages over time was further examined. After a 2 h incubation of macrophages containing bacteria, approximately half of the wild-type *K. pneumoniae* were viable in the macrophages, while more than 90% of the *crr* mutant survived in the macrophages. For the wild-type, *crr* mutant and *crr* complemented strains, the survival rates of intracellular bacteria gradually decreased. However, the survival rates of the *crr* mutant remained higher than those of the other strains during 16 h of incubation (Figure 4C). The results suggested that deficiency of *crr* in *K. pneumoniae* increased the resistance of bacteria to phagocytosis by reducing macrophage ingestion and degradation inside macrophages.

### 3.4. Deficiency of crr Resulted in Overexpression of Another EII Complex, EtcABC

We previously reported that overexpression of KPN00353-KPN00352-KPN00351, homologs of EtcABC, increased the production of CPS in *K. pneumoniae* [31]. In addition, the CPS production of the *etcABC* mutant was not significantly different from that of the wild-type (average CPS amount of *etcABC* mutant and wild-type was 0.060 mg/L and 0.061 mg/mL, respectively, and Figure 3C). Therefore, we compared the transcription levels of *etcABC* in the *crr* mutant and wild-type. As observed by RT-qPCR, the amounts of *etcA*, *etcB* and *etcC* mRNAs in the *crr* mutant were higher than those in the wild-type (Figure 5A), indicating that deficiency of *crr* may activate the transcription of *etcABC*. To elucidate the role of *etcABC* in the *crr* mutant, *etcABC* was deleted in the *crr* mutant. By quantifying the CPS, we determined that the CPS amount in the Δ*crr/*Δ*etcABC* double mutant was lower than that in the *crr* mutant but not as low as that in the wild-type. Complementation of *etcABC* in the Δ*crr/*Δ*etcABC* double mutant restored the CPS amount close to that in the *crr* mutant (Figure 5B). Since we observed that genes in the *cps* cluster (*galF*, *wzi* and *gnd*) were affected by *crr* (Figure 3D,E), the transcriptional activity of *galF* in the Δ*crr/*Δ*etcABC* double mutant was detected by RT-qPCR using *recA* as a reference. The results demonstrated that the mRNA levels of *galF* in the Δ*crr/*Δ*etcABC* double mutant were lower than those in the *crr* mutant but were increased by complementation of *etcABC* in the *crr* mutant (Figure 5C), indicating that CPS production in *K. pneumoniae* could be affected by Crr via EtcABC overexpression.

## 4. Discussion

The *crr* gene encodes Crr, which is also known as IIA^Glc^ or EIIA^Glc^ and was named IIIGlc in early literature [41,42]. This study is the first report about the roles of Crr in regulating bacterial capsule formation in *K. pneumoniae* and resistance to phagocytosis. Deficiency of *crr* resulted in an increase in CPS production by elevating the transcriptional activities of *cps*-related genes in *K. pneumoniae* (Figure 3). Deficiency of *crr* in *K. pneumoniae* also enhanced bacterial resistance to hydrogen peroxide-induced oxidative stress (Figure 4A). Deletion of *crr* in *K. pneumoniae* helped bacteria resist engulfment by macrophages (Figure 4B,D) and bactericidal activity in phagosomes (Figure 4C). Moreover, we showed that dual EII systems, Crr and EtcABC, affected CPS production in *K. pneumoniae* (Figure 5C). Among these two PTSs, Crr negatively affected the transcriptional activity of *etcABC* (Figure 5A). Figure 6 summarizes the regulatory control of Crr on *etcABC*, *galF*, *wzi*, and *gnd* to affect CPS production in this study.

Although capsule was reported to protect the yeast *Cryptococcus neoformans* from killing by hydrogen peroxide [19], the relationship between bacterial CPS and bacterial resistance to hydrogen peroxide is unclear. Ares et al. demonstrated that capsule deficiency in *K. pneumoniae* dramatically enhanced macrophage-mediated phagocytosis [43]. Cano et al. also found a similar phenomenon: a CPS mutant was engulfed by macrophages in higher numbers than its parental strain. However, Cano et al. further demonstrated that the capsule is dispensable for *Klebsiella* survival in macrophages when bacteria are engulfed via a route other than opsonization. They also showed that *K. pneumoniae* prevented the fusion of lysosomes to the endosome containing itself in macrophages [44]. Therefore, we speculated that the increase in CPS was just one of the factors mediated by the *crr* mutant to resist phagocytosis. Deficiency of *crr* may affect some factors to impede lysosome fusion and reduce oxidative stress in phagocytes.

There are several reports about one PTS regulating another, but there is no clear mechanism. For example, Heravi et al. showed that there is cross talk between the noncognate EIIA and EIIB domains in the PTS in *Bacillus subtilis* [45]. Chaillou et al. demonstrated that the EIIB^man^ of *Lactobacillus pentosus* decreased the rate of PEP-dependent phosphorylation of fructose, indicating that mannose PTS expression modulated the fructose-specific PTS. However, the kind of fructose-specific component regulated by EIIB^man^ has not been confirmed [46]. We previously demonstrated that overexpression of EtcA-EtcB-EtcC homologs, KPN00353-KPN00352-KPN00351, enhanced CPS production in *K. pneumoniae* [31]. In this study, we showed that deletion of *crr* in *K. pneumoniae* enhanced the transcriptional activity of *etcABC* (Figure 5A). This is the first report of one PTS regulating the transcription of another PTS. As there is no DNA-binding domain in Crr, an additional unknown factor is the molecular mechanism mediated by Crr to affect *etcABC*.

Crr and EtcABC are not transcriptional factors so a potential factor may be involved in the regulation of *galF*, *wzi* and *gnd* by Crr or EtcABC. In the present study, we did not find this potential factor. We predict that Crr or EtcABC may interact with a potential transcriptional factor, and then change its activity, modulating the transcription of *galF*, *wzi* and *gnd* eventually. Although cAMP and CRP were reported to repress CPS biosynthesis in *K. pneumoniae* CG43S3 [47], we did not find this phenomenon in *K. pneumoniae* STU1 and clinical strains. First, deficiency of *crp* in *K. pneumoniae* STU1 did not show much longer viscous strings and more CPS than wild-type (our preliminary data). Second, deletion of *crr* decreased cAMP production, but overexpression of *etcABC* greatly increased the cAMP levels in the *crr* mutant and Δ*crr/*Δ*etcABC* double mutant in our previous study [32]. In this study, deletion of *crr* increased the transcriptional levels of genes in the *cps* locus (Figure 3D and Figure 5C). Complementation of *etcABC* in the Δ*crr/*Δ*etcABC* double mutant restored the transcriptional levels of genes in the *cps* locus to those of *cps* genes in the *crr* mutant (Figure 5C). In brief, both *crr* and *etcABC* had positive effects on cAMP production, but *crr* had effects on CPS production different from those of *etcABC*. Therefore, we speculate that CRP-cAMP may not repress the CPS production in *K. pneumoniae*, which was mentioned by Lin et al. [47]. The effects of CRP-cAMP on CPS production in *K. pneumoniae* STU1 are being studied. Furthermore, we inferred that all or one of the components of EtcABC may interact with an additional protein to activate transcription of the *cps* genes, leading to an increase in CPS production in *K. pneumoniae*.

In this study, we found that the transcriptional activities of *etcABC* were all increased in the *crr* mutant (Figure 5A). The phosphorylation state of EIIA is important for its activity [27,48]. Overexpression of EtcABC or just one protein of this complex may change the phosphorylation state of EtcA or EtcB. However, the phosphorylation states of EtcA and EtcB were not confirmed because the usage of radioactive chemicals was limited in our laboratory. Therefore, we overexpressed all of the EtcABC components, not just one of them, in *K. pneumoniae* to study why the *crr* mutant displayed increased CPS production in this study. The role of each component of EtcABC in CPS synthesis should be studied in the future, including the phosphorylation states of EtcA and EtcB.

In addition to this study, only one report at present has been on the roles of Crr in *K. pneumoniae*. Oh et al. demonstrated that the production of 1,3-propanediol by a *K. pneumoniae crr* mutant grown in medium containing glycerol and glucose was enhanced compared to that by the parent strain. However, the production of 1,3-propanediol by the *K. pneumoniae crr* mutant and its parent strain grown in medium containing glycerol without glucose was not different, indicating that Crr was involved in the regulation of 1,3-propanediol production by glucose [29]. Moreover, Lee et al. showed that a high glucose concentration (0.5%) enhanced CPS biosynthesis and *cps* gene expression in a *K. pneumoniae* clinical isolate that was isolated from a patient with pyogenic liver abscess (PLA) and endophthalmitis. Their clinical retrospective study found that poor glycemic control in diabetic patients increased the risk of *K. pneumoniae*-mediated invasive syndrome [49]. From these studies, we speculated that Crr and EtcABC may play roles in *K. pneumoniae* infection, especially for diabetic patients with inadequate glycemic control.

Although *K. pneumoniae* and *E. coli* are members of *Enterobacteriaceae* and *K. pneumoniae* Crr is similar to *E. coli* Crr, *etcABC* is unique in *K. pneumoniae* and lacking in *E. coli* [30]. The mode of regulation of *cps* genes by Crr via *etcABC* may be a distinct molecular mechanism in *K. pneumoniae* pathogenicity. In addition, although Crr is glucose-specific EIIA in *E. coli* [26], we found that the growth of the *K. pneumoniae crr* mutant did not show defects in a medium containing glucose as sole carbon source compared to that of the wild-type (our preliminary data). Therefore, the cognate sugars of Crr and EtcABC in *K. pneumoniae* and the connection of these cognate sugars and CPS need to be investigated in the future. This study provided the view that one member of the PTS, Crr, repressed the transcriptional activity of another PTS member, EtcABC, to reduce the transcriptional activity of *cps* genes and CPS production in *K. pneumoniae*. Deficiency of *crr* in *K. pneumoniae* rendered bacteria more resistant to phagocytosis, including more resistance to engulfment by macrophages and survival in macrophages.

## 5. Conclusions

In addition to transporting and phosphorylating sugars from the environment, bacterial PTS components play various roles in bacterial physiology. The glucose-specific enzyme IIA (Crr) is common in many enteric bacteria, while EtcABC, an enzyme II (EII) complex specific to unknown sugars, is unique in *K. pneumoniae*. This study demonstrated that these two different EII systems affected the transcriptional activities of *cps*-related genes and CPS production, which is an essential virulence factor for *K. pneumoniae* infection. In addition to the effects on CPS, inactivation of glucose-specific enzyme IIA (Crr) made *K. pneumoniae* more resistant to hydrogen peroxide-induced oxidative stress, macrophage engulfment and bactericidal effects inside macrophages. Moreover, the regulation of the transcriptional activities of one EII complex by another PTS component suggested a complicated bacterial response to environmental carbon source fluctuations.

## Figures and Tables

**Figure 1 microorganisms-09-00335-f001:**
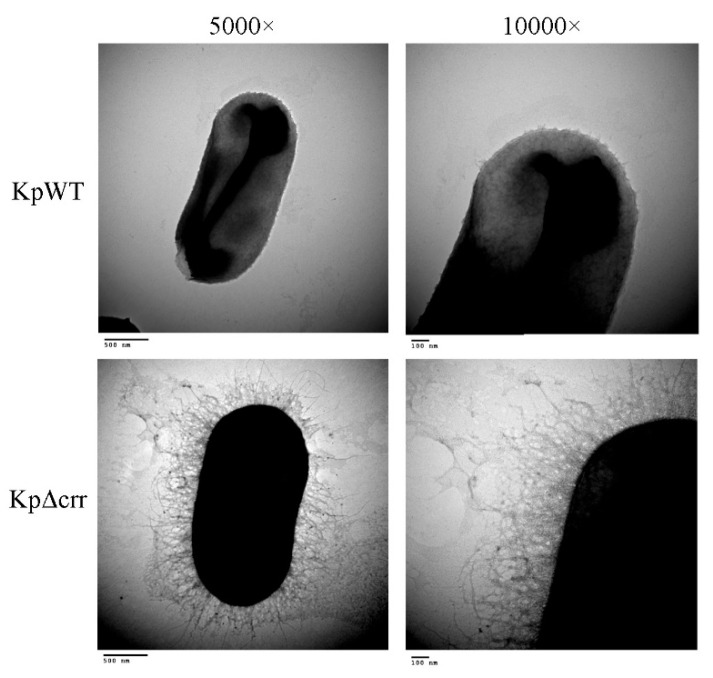
Phenotypic characteristics of the *crr* mutant in *Klebsiella pneumoniae* STU1. Representative TEM images from three independent experiments. KpWT: *K. pneumoniae* STU1 wild-type. KpΔcrr: *K. pneumoniae crr* mutant.

**Figure 2 microorganisms-09-00335-f002:**
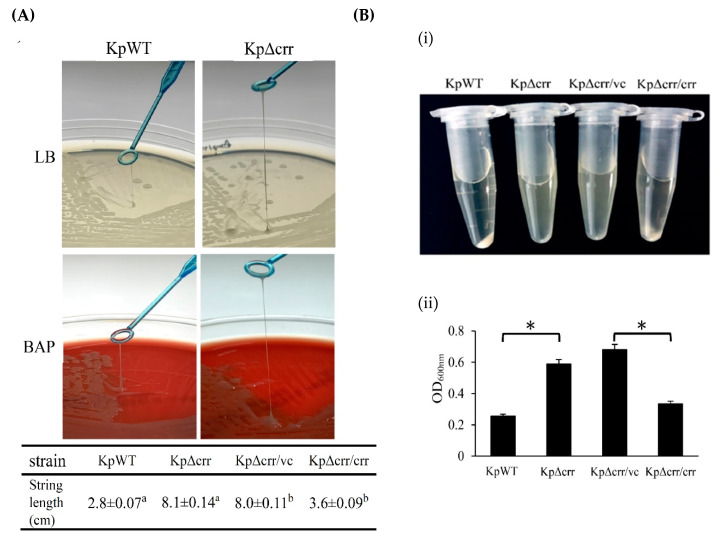
Deficiency of *crr* enhanced the mucoviscosity of *K. pneumoniae*. (**A**) Hypermucoviscosity phenotype observed by the string test. LB: LB agar plate. BAP: 5% sheep blood agar plate. (**B**) Bacterial mucoviscosity assay. (i) Image of bacterial culture after centrifugation. (ii) The absorbance of the supernatant was determined at 600 nm using a spectrophotometer. KpWT: *K. pneumoniae* STU1 wild-type. KpΔcrr: *K. pneumoniae crr* mutant. KpΔcrr/vc: *K. pneumoniae crr* mutant carrying pBAD33 (vector control). KpΔcrr/crr: *K. pneumoniae crr* mutant carrying pBAD33::crr (*crr* complemented strain). The superscript a and b in (**A**) and asterisk (∗) in (**B**) represent *p* < 0.05.

**Figure 3 microorganisms-09-00335-f003:**
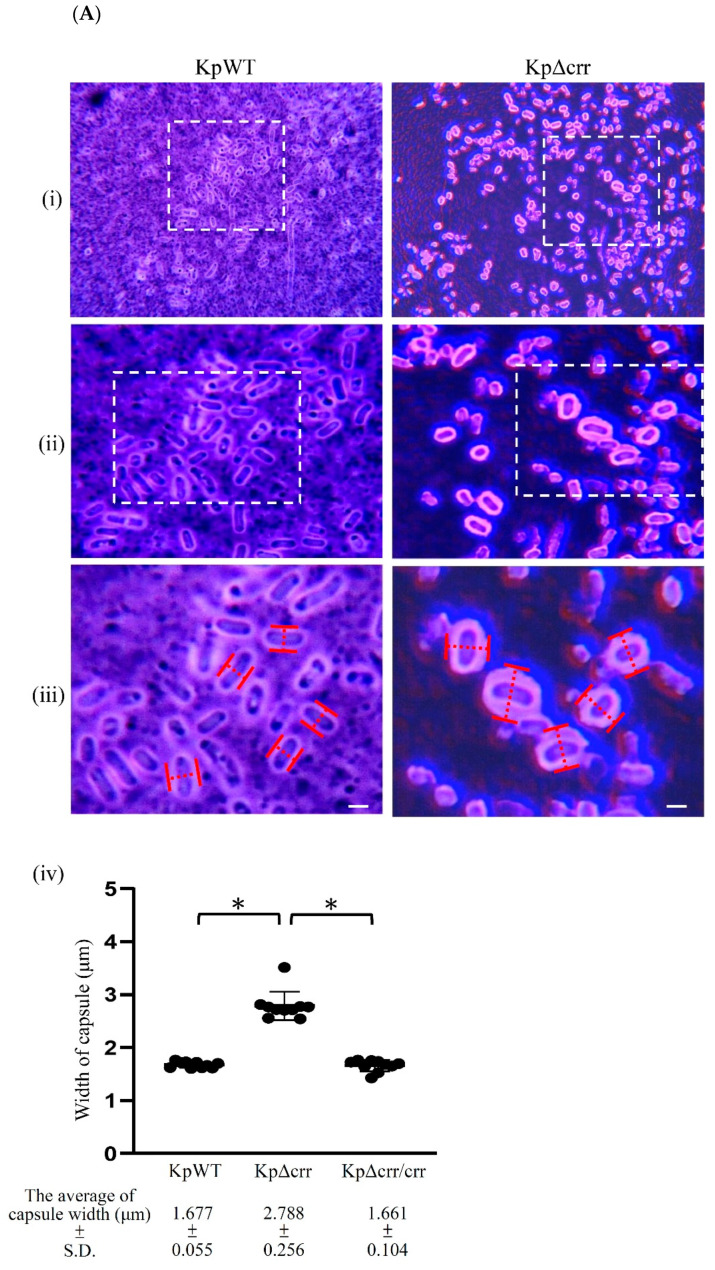
Deficiency of *crr* enhanced CPS production in *K. pneumoniae*. (**A**) Light microscope images of bacteria after India ink staining. Capsular saccharide was visualized by a cleared zone (ink exclusion) around the bacteria. (**i**) Images were obtained at 400 × magnification. (**ii**) Enlargement of the rectangle outlined with a white dashed line in (**i**). (**iii**) Enlargement of the rectangle outlined with a white dashed line in (**ii**). (**iv**) The width of the capsule was analyzed by measuring the red dashed line in (**iii**). The scale bar, white horizontal line, in subfigure (**iii**) = 1 µm. (**B**) Representative images of bacteria on Congo red agar (CRA) plates. (**C**) Quantification of bacterial CPS. (**D**,**E**) Transcriptional analysis of the housekeeping gene *recA* and *cps* locus genes, including *galF*, *wzi* and *gnd*, by dye-based RT-qPCR. The relative mRNA levels indicate that the mRNA amount in KpΔcrr was normalized to that in KpWT in (**D**) or that the mRNA amount in KpΔcrr/crr was normalized to that in KpΔcrr/vc in (**E**). KpWT: *K. pneumoniae* STU1 wild-type. KpΔcrr: *K. pneumoniae crr* mutant. KpΔcrr/crr: *K. pneumoniae crr* mutant carrying pBAD33::crr (*crr* complemented strain). KpΔcrr/vc: *K. pneumoniae* crr mutant carrying pBAD33 (vector control). The photos and data regarding KpΔcrr could be used to represent those of KpΔcrr/vc in (**A**,**B**,**C**) because of their similar phenotypes. The asterisk (∗) represents *p* < 0.05.

**Figure 4 microorganisms-09-00335-f004:**
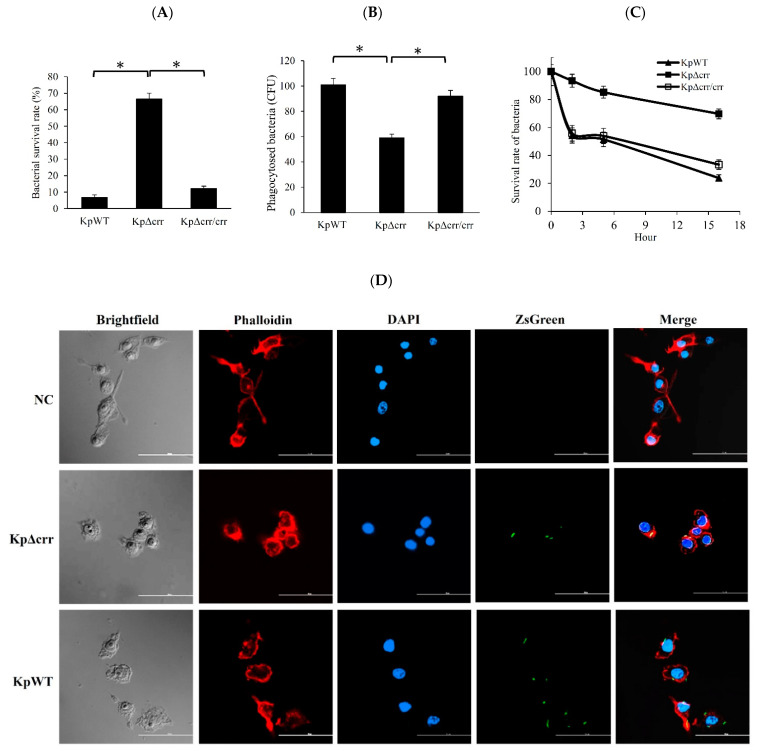
The effect of *crr* on resistance to hydrogen peroxide, macrophage ingestion and bactericidal activity in macrophages. (**A**) Survival rate of bacteria exposed to 1% hydrogen peroxide for 30 min. The survival rate of each strain was calculated by dividing the CFU of bacteria with H_2_O_2_ treatment by the CFU of bacteria without H_2_O_2_ treatment. (**B**) Evaluation of bacteria ingested by macrophages. After 1 h of infection and 30 min of antibiotic treatment, the macrophages were lysed and plated to calculate the recovery of intracellular bacteria. Values shown are the mean CFU of viable bacteria per well ± SD of three independent experiments. (**C**) Survival rate of intracellular bacteria resistant to phagocytosis by macrophages after 1 h of infection, 30 min of antibiotic treatment and incubation for 2, 5 or 16 h. The survival rate of each strain was calculated by dividing the CFU of viable bacteria after incubation by the CFU of viable bacteria without incubation (0 h). (**D**) Internalization of bacteria by macrophages observed by CLSM. The macrophages were infected for 1 h with bacteria carrying pBSK-Km::ZsGreen (green). After 30 min of antibiotic treatment, the cells were visualized by staining F-actin with phalloidin (red). The cell nuclei were stained with DAPI (blue). Representative 2D images were acquired by choosing the maximum intensity projection from 3D CLSM images in three different experiments. Scale bar (white horizontal line in each subfigure) = 50 µm. NC: no bacteria, as a negative control. KpWT: *K. pneumoniae* STU1 wild-type. KpΔcrr: *K. pneumoniae crr* mutant. KpΔcrr/crr: *K. pneumoniae crr* mutant carrying pBAD33: crr (*crr* complemented strain). The data for the *crr* mutant could be used to represent that of the *K. pneumoniae crr* mutant carrying pBAD33 (vector control) in (**A**–**C**) because of their similar results. The asterisk (∗) in (**A**) and (**B**) represents *p* < 0.05. (**C**) The *p*-values of comparing the data from KpWT and KpΔcrr at second hour, 5th or 16th hour were less than 0.05.

**Figure 5 microorganisms-09-00335-f005:**
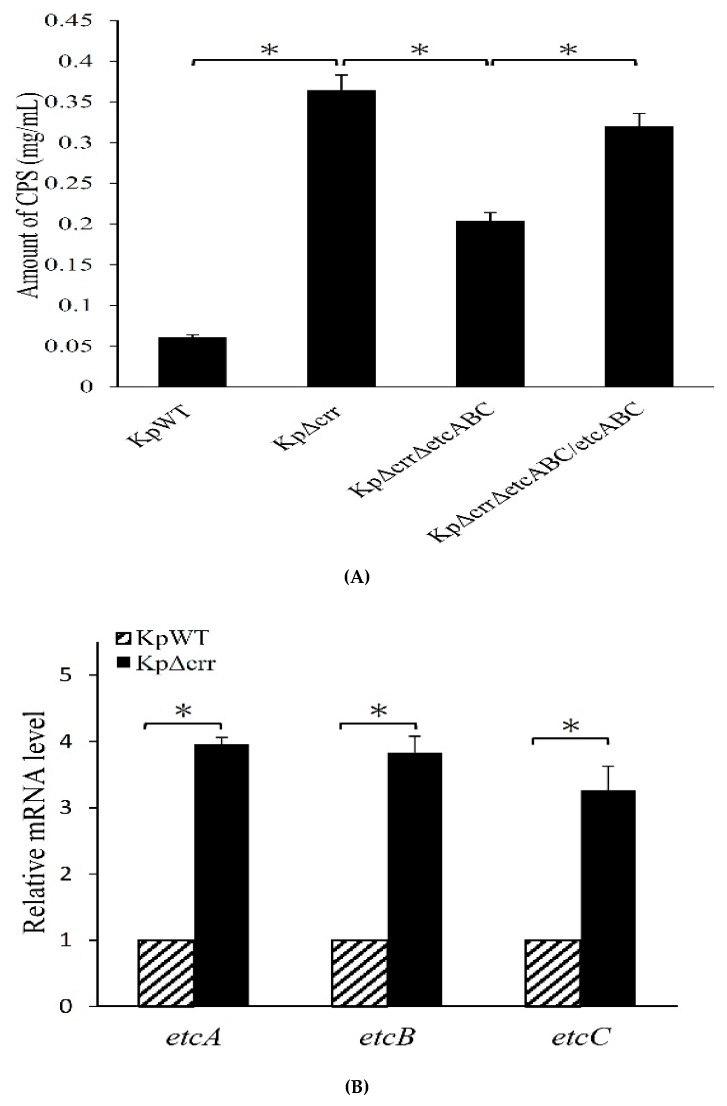
Overexpression of *etcABC* increased CPS production in *K. pneumoniae*. (**A**) Transcriptional analysis of *etcA*, *etcB* and *etcC* in the wild-type (slashed bar) and *crr* mutant (black bar) by probe-based RT-qPCR. The relative mRNA levels indicate that the mRNA amount in KpΔcrr was normalized to that in KpWT. (**B**) Quantification of CPS. KpWT: *K. pneumoniae* STU1 wild-type. KpΔcrr: *K. pneumoniae crr* mutant. KpΔcrrΔetcABC: both *crr* and *etcABC* were deleted in *K. pneumoniae*. KpΔcrrΔetcABC/etcABC: KpΔcrrΔetcABC carrying pBSK-Gm: Km::etcABC for overexpression of *etcABC*. (**C**) Transcriptional analysis of the housekeeping gene *recA* and *cps* locus gene, *galF*, by probe-based RT-qPCR. The relative mRNA levels indicate that the mRNA amount in each strain was normalized to that in KpWT. KpWT (black bar): *K. pneumoniae* STU1 wild-type. KpΔcrr (white bar): *K. pneumoniae crr* mutant. KpΔetcABC (slashed bar): *K. pneumoniae etcABC* mutant. KpΔcrrΔetcABC/vc (dotted bar): *K. pneumoniae* Δ*crr*/Δ*etcABC* double mutant carrying pBSK-Gm: Km as a vector control. KpΔcrrΔetcABC/etcABC (gray bar): *K. pneumoniae* Δ*crr*/Δ*etcABC* double mutant carrying pBSK-Gm: Km::etcABC to overexpress *etcABC*. The asterisk (∗) represents *p* < 0.05.

**Figure 6 microorganisms-09-00335-f006:**
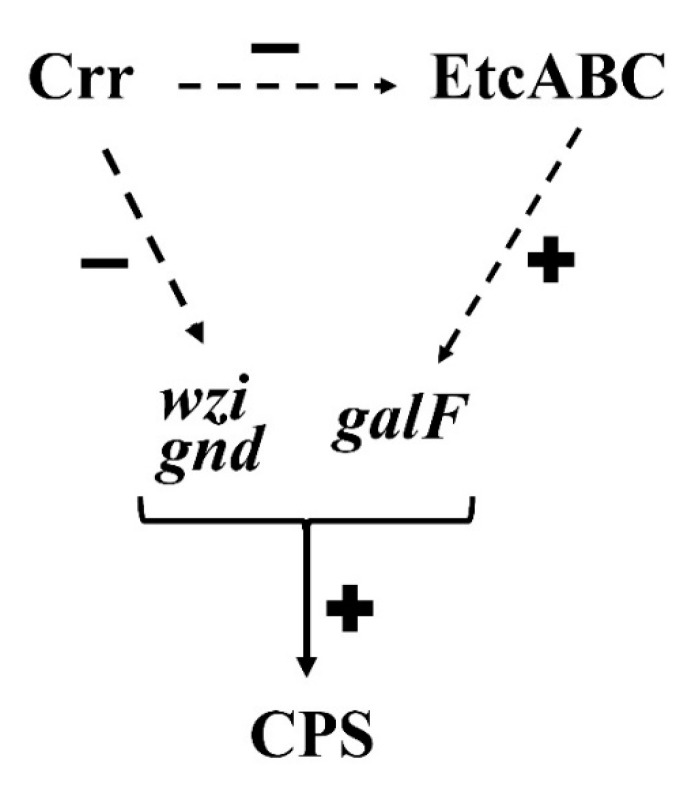
Model for the roles of Crr and EtcABC in the regulation of CPS synthesis in *K. pneumoniae*. Crr affects activity of EtcABC negatively but EtcABC affects the transcriptional activity of *galF* positively. In addition, Crr affects transcriptional activity of *wzi* and *gnd* negatively. The roles of *galF*, *wzi* and *gnd* are involved in CPS synthesis in *K. pneumoniae*. (+) and (−) represent positive and negative regulation respectively. Arrows with dashed line mean indirect regulation.

## Data Availability

Not applicable.

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
