# Peer review of "The PTS Components in Klebsiella pneumoniae Affect Bacterial Capsular Polysaccharide Production and Macrophage Phagocytosis Resistance"

_microorganisms, 2021, doi:10.3390/microorganisms9020335_

Round 1
Reviewer 1 Report
The aim is stated clear. The authors stated clearly what study found and how they did it. The title is informative and relevant.
The references are relevant and recent. The cited sources are referenced correctly. Appropriate and key studies are included.
The introduction reveals what is already known about this topic. The research question is clearly outlined. The research question also justified given what is already known about the topic.
The study methods are valid and reliable. There are enough details provided in order to replicate the study.
The data is presented in an appropriate way. T
The text in the results adds to the data and it is not repetitive. Statistically significant results are clear. It is clear which results are with practical meaning. Results are discussed from different angles and placed into context without being over-interpreted.
The conclusions answer the aim of the study. The conclusions are supported by references and own results.
The limitations of the study are not fatal, but they are opportunities to inform future research.
The study design is appropriate to answer the aim. This study added to what is already in the topic.
The article is consistent within itself.
There are not major flaws of this article.
Overall strengths of the article are the various metholodoly and the valid result.
Specific comments on weaknesses of the article and what could be improved:
Major points - none
Minor points
- Please, state the limitation of the study.
- Would you suggest the implementation of the findings in the clinical settings?
Author Response
Reviewer1
- Please, state the limitation of the study.
Response 1-1:
“Crr and EtcABC are not transcriptional factors so a potential factor may be involved in the regulation of galF, wzi and gnd by Crr or EtcABC. In the present study, we have not found this potential factor. We predict that Crr or EtcABC may interact with a potential transcriptional factor, and then change its activity, modulating the transcription of galF, wzi and gnd eventually.”
/ The discussion above is written on line 432-436 in the revised manuscript.
- Would you suggest the implementation of the findings in the clinical settings
Response 1-2:
We have not known cognate sugars of Crr and EtcABC. We speculate that the effects of environmental sugars on Klebsiella infection may be related to Crr and EtcABC. However, we think the implementation of the findings in the clinical settings is appropriate after cognate sugars of Crr and EtcABC are confirmed. Therefore, we do not recommend the findings in the clinical settings in this revised manuscript.
Reviewer 2 Report
In this manuscript, Panjaitan et al. demonstrated that Crr, a IIA enzyme, controls production of capsular polysaccharide (CPS) in K. pneumoniae. The authors show that Crr exerts control at a regulatory level by acting a repressor of the EtcABC PTS system which positively controls CPS. The results shown by the author support their hypothesis and conclusions. Only, a few minor comments are listed below, which would increase the impact of the manuscript:
Comments
- Line 301-302: ‘which was…strains.’ Are authors trying to say that the strings in crr mutant was 5 mm longer than wt? This would flow better as a separate sentence.
- Line 339-341: ‘In contrast….vector control (Figure 3E).’ Gene expression levels in all strains should be shown together in a single panel for better comparison.
- Line 394-395: ‘The results….crr mutant’ Did the authors also examine the expression levels of wzi and gnd in these strains? This would further support the hypothesis stated in lines 397-398.
- A figure depicting the regulatory control of Crr on etcABC, galF, wzi and gnd to affect CPS production would greatly benefit the manuscript.
Author Response
Reviewer2
1. Line 301-302: ‘which was…strains.’ Are authors trying to say that the strings in crr mutant was 5 mm longer than wt? This would flow better as a separate sentence.
Response 1:
The length of string in the wild-type was approximately 2.8 cm and length of string in crr mutant was 8.1 cm (line 295-296 in revised manuscript). Therefore, length of string in the crr mutant was 53 mm (NOT 5 mm) longer than that in wt. However, the length of string in each strain in this study was more than 5 mm, indicating that they were all hypermucoviscous strains (line 296-297 in the revised manuscript.). Therefore, we did not add the sentence in the revised manuscript which the reviewer mentioned.
2. Line 339-341: ‘In contrast….vector control (Figure 3E).’ Gene expression levels in all strains should be shown together in a single panel for better comparison.
Response 2:
In figure 3D, the relative transcription levels of genes in the crr mutant were compared to those in wild-type. Therefore, the values of gene transcription levels in wild-type were defined as one in figure 3D. In figure 3E, the relative transcription levels of genes in the crr complemented strain were compared to those in vector control. Therefore, the values of gene transcription levels in vector control were defined as one in figure 3E. The strains in figure 3E have plasmids but the strains in figure 3D do not. Because criteria in figure 3D and 3E are different, we do not show data from these four strains in a single panel in the revised manuscript.
3. Line 394-395: ‘The results….crr mutant’ Did the authors also examine the expression levels of wzi and gnd in these strains? This would further support the hypothesis stated in lines 397-398.
Response 3:
We have examined the transcriptional levels of wzi and gnd in â–³crrâ–³etcABC/vc and â–³crrâ–³etcABC/etcABC as reviewer’s suggestion. However, the results did not show significant different transcriptional levels of wzi and gnd in â–³crrâ–³etcABC/vc and â–³crrâ–³etcABC/etcABC. We think that EtcABC affects the transcription level of galF but not those of wzi and gnd.
4. A figure depicting the regulatory control of Crr on etcABC, galF, wzi and gnd to affect CPS production would greatly benefit the manuscript.
Response 4:
We have given Figure 6 to summarize the finding in this study as reviewer’s suggestion. The discussion about Figure 6 is written on line 403-405 in the revised manuscript.